# Pilot study evaluating everolimus molecular mechanisms in tuberous sclerosis complex and focal cortical dysplasia

**Dominique F. Leitner**[1], **Evgeny Kanshin**[2], **Manor Askenazi**[3,4], **Yik Siu**[5], **Daniel Friedman**[1], **Sasha Devore**[1], **Drew Jones**[5], **Beatrix Ueberheide**[2,4,6], **Thomas Wisniewski**[6,7,8], **Orrin Devinsky**[1]*

1 Comprehensive Epilepsy Center, New York University School of Medicine, New York, New York, United States of America, 2 Proteomics Laboratory, Division of Advanced Research Technologies, NYU School of Medicine, New York, New York, United States of America, 3 Biomedical Hosting LLC, Arlington, Massachusetts, United States of America, 4 Department of Biochemistry and Molecular Pharmacology, New York University School of Medicine, New York, New York, United States of America, 5 Metabolomics Core Resource Laboratory, New York University School of Medicine, New York, New York, United States of America, 6 Center for Cognitive Neurology, Department of Neurology, New York University School of Medicine, New York, New York, United States of America, 7 Department of Psychiatry, New York University School of Medicine, New York, New York, United States of America, 8 Department of Pathology, New York University School of Medicine, New York, New York, United States of America

* Orrin.Devinsky@nyulangone.org

**Data Availability Statement:** All relevant data are within the paper and its Supporting information files.

## Abstract

### Background

Tuberous sclerosis complex (TSC) and some focal cortical dysplasias (FCDs) are associated with dysfunctional mTOR signaling, resulting in increased cell growth and ribosomal S6 protein phosphorylation (phospho-S6). mTOR inhibitors can reduce TSC tumor growth and seizure frequency, and preclinical FCD studies indicate seizure suppression. This pilot study evaluated safety of mTOR inhibitor everolimus in treatment resistant (failure of >2 anti-seizure medications) TSC and FCD patients undergoing surgical resection and to assess mTOR signaling and molecular pathways.

### Methods and findings

We evaluated everolimus in 14 treatment resistant epilepsy patients undergoing surgical resection (4.5 mg/m² daily for 7 days; n = 4 Active, mean age 18.3 years, range 4–26; n = 10, Control, mean age 13.1, range 3–45). Everolimus was well tolerated. Mean plasma everolimus in Active participants were in target range (12.4 ng/ml). Brain phospho-S6 was similar in Active and Control participants with a lower trend in Active participants, with Ser235/236 1.19-fold (p = 0.67) and Ser240/244 1.15-fold lower (p = 0.66). Histologically, Ser235/236 was 1.56-fold (p = 0.37) and Ser240/244 was 5.55-fold lower (p = 0.22). Brain proteomics identified 11 proteins at <15% false discovery rate associated with coagulation system (p = 1.45x10⁻⁹) and acute phase response (p = 1.23x10⁻⁶) activation. A weighted gene correlation network analysis (WGCNA) of brain proteomics and phospho-S6 identified 5 significant modules. Higher phospho-S6 correlated negatively with cellular respiration and synaptic transmission and positively with organophosphate metabolic process, nuclear

**Funding:** Funding was provided by Finding A Cure for Epilepsy and Seizures (FACES, http://faces.med. nyu.edu/) for supplies, reagents, and salary support to DL, OD, National Institutes of Health (NIH, https://www.nih.gov/) P01AG060882 for salary support to TW, and Novartis (https://www. novartis.com/) provided partial funding support as well as the everolimus used in the study. The funders had no role in study design, data collection and analysis, decision to publish, or preparation of the manuscript.

**Competing interests:** I have read the journal's policy and the authors of this manuscript have the following competing interests: All the authors report no conflicts of interest. Below are the disclosures of the three authors: Daniel Friedman receives salary support for consulting and clinical trial related activities performed on behalf of The Epilepsy Study Consortium, a non-profit organization. Dr. Friedman receives no personal income for these activities. NYU receives a fixed amount from the Epilepsy Study Consortium towards Dr. Friedman's salary. Within the past two years, The Epilepsy Study Consortium received payments for research services performed by Dr. Friedman from: Alterity, Baergic, Biogen, BioXcell, Cerevel, Cerebral, Jannsen, Lundbeck, Neurocrine, SK Life Science, and Xenon. He has also served as a paid consultant for Neurelis Pharmaceuticals and Receptor Life Sciences. He has received research support from NINDS, CDC, Epitel, and Neuropace unrelated to this study. He holds equity interests in Neuroview Technology. He received royalty income from Oxford University Press. Sasha Devore receives salary support from the National Institutes of Health, Department of Defense, and the Templeton World Charity Foundation unrelated to this study. Orrin Devinsky receives grant support from NINDS, NIMH, MURI, CDC and NSF. He has equity and/or compensation from the following companies: Tilray, Receptor Life Sciences, Qstate Biosciences, Tevard, Empatica, Engage, Egg Rock/ Papa & Barkley, Rettco, SilverSpike, and California Cannabis Enterprises (CCE). He has received consulting fees from Zogenix, Ultragenyx, BridgeBio, and Marinus. He holds patents for the use of cannabidiol in treating neurological disorders but these are owned by GW Pharmaceuticals and he has waived any financial interests. He holds other patents in molecular biology. This does not alter our adherence to PLOS ONE policies on sharing data and materials.

mRNA catabolic process, and neuron ensheathment. Brain metabolomics identified 14 increased features in Active participants, including N-acetylaspartylglutamic acid. Plasma proteomics and cytokine analyses revealed no differences.

## Conclusions

Short-term everolimus before epilepsy surgery in TSC and FCD resulted in no adverse events and trending lower mTOR signaling (phospho-S6). Future studies should evaluate implications of our findings, including coagulation system activation and everolimus efficacy in FCD, in larger studies with long-term treatment to better understand molecular and clinical effects.

## Clinical trials registration

ClinicalTrials.gov NCT02451696.

## Introduction

In patients with tuberous sclerosis complex (TSC) and in some focal cortical dysplasia (FCD) cases, treatment resistant epilepsy (TRE) is associated with increased activation of mammalian target of rapamycin (mTOR) signaling [1, 2]. TSC is characterized primarily by non-malignant tumors in multiple organ systems as a result of mutations in *TSC1* (hamartin) or *TSC2* (tuberin), which normally combine to downregulate mTOR signaling related cell growth and proliferation [3, 4]. FCD is focally malformed cortex with aberrant lamination and dysmorphic neurons, and associated with epileptiform activity and seizures [5]. FCD may result from somatic mosaicism with pathogenic gene variants in the mTOR signaling pathway, particularly FCD type II with dysmorphic neurons [6, 7]. Dysmorphic and balloon neurons are found histologically in TSC and FCD, with high ribosomal protein S6 phosphorylation (phospho-S6) due to increased mTOR pathway activation [6, 8].

mTOR inhibitors, including everolimus and the related rapamycin/sirolimus [9–11], treat mTORopathies by binding to upstream FKBP12 (encoded by *FKBP1A*) and inhibiting mTOR signaling [11]. Everolimus reduced brain phospho-S6 in TSC mouse models [12, 13]. The EXIST (EXamining everolimus In a Study of TSC) clinical trials led to FDA approval to reduce subependymal giant cell astrocytomas (SEGAs) and seizures [14, 15]. In FCD genetic models, rapamycin reduced seizure frequency and migrational/morphological defects [1]. Short-term rapamycin for 7 days in glioblastoma patients reduced phospho-S6 [16]. Two clinical studies are evaluating everolimus in FCD, the current study and a pending Korean study assessing seizure frequency (NCT03198949). The effect of everolimus on brain phospho-S6 and molecular mechanisms in TSC and FCD patients with TRE undergoing surgical resection has not been studied.

In this pilot phase 2 study, we evaluated the safety and molecular mechanisms of short-term everolimus (7 days) in TSC and FCD patients with TRE.

## Materials and methods

### Participants

This pilot study was registered at clinicaltrials.gov (NCT02451696) and approved by the New York University School of Medicine Institutional Review Board (IRB; s14-00245). Recruitment

**Table 1. Case history.**

| Case ID | NP Group | Age | Sex | Brain Region | Neuropathology Diagnoses | Plasma Evero. (ng/ml) |
|---|---|---|---|---|---|---|
| *Control* | | | | | | |
| NYU-C-001 | FCD | 8 | F | TL | oligodendroglioma, WHO grade II; focal hypercellularity; reactive gliosis | < LOD |
| NYU-C-002 | FCD | 16 | M | FR | cortical dysplasia; microglial activation; gliosis | < LOD |
| NYU-C-003 | FCD | 11 | F | TL | cortical dysplasia; white matter hypercellularity; gliosis; heterotopia; microglial activation | < LOD |
| NYU-C-004 | TSC | 3 | F | PA | dysplastic cortex; gliosis; abundant balloon cells | < LOD |
| NYU-C-005 | FCD | 3 | F | TL | reactive gliosis; focal cortical disorganization | < LOD |
| NYU-C-006 | TSC | 4 | F | PA | cortical dysplasia; balloon cells; white matter hypercellularity; gliosis; heterotopia; microglial activation; perivascular lympho-histiocytic infiltrate; mineralization | < LOD |
| NYU-C-007* | FCD | 11 | M | FR | FCD IIB | < LOD |
| NYU-C-008* | TSC | 13 | M | TL | balloon cells; cortical dysplasia; heterotopia; white matter hypercellularity; gliosis; microglial activation; mineralization; perivascular inflammation | < LOD |
| NYU-C-009 | FCD | 45 | M | TL | focal dysplasia; focal neuronal loss; reactive gliosis | < LOD |
| NYU-C-010 | FCD | 17 | F | TL | FCD IB | < LOD |
| *Active* | | | | | | |
| NYU-002 | FCD | 18 | M | FR | FCD IB | 19.7 |
| NYU-003 | TSC | 4 | F | FR | neuronal disorganization; dysplastic neurons | 7.1 |
| NYU-004 | TSC | 25 | F | TL | dysembryoplastic neuroepithelial tumor, WHO grade I | 10.2 |
| NYU-005 | FCD | 26 | F | TL | heterotopia; focal dysplasia; reactive gliosis | 12.4 |

*no plasma available for some assays.

LOD = limit of detection; TL = temporal lobe; FR = frontal lobe; PA = parietal lobe; NP = neuropathology.

began in 2014 and closed in 2019. Written informed consent for participants was provided by the parent/legal guardian/health care proxy, as the TSC and FCD patients were considered cognitively impaired. Male and female participants were included from ages 2 to 50 years of age, with a diagnosis of TRE (as defined by failure of > 2 anti-seizure medications [ASMs]), and undergoing routine surgical resection for clinically diagnosed TSC [17] or MRI-diagnosed FCD. Patients were excluded if using strong CYP3A4 inhibitors and more than one strong CYP3A4 inducer (e.g., phenytoin, carbamazepine, phenobarbital). Fifteen participants were enrolled. One participant developed fever on day 6 of study treatment but investigator deemed it not related to study drug; however, the participant withdrew from the study, leaving a total of 14 participants. Participants were age (Active: 18.3 years mean, range 4–26, Control: 13.1 years, range 3–45; p = 0.48) and sex matched (p>0.99). Clinical histories are detailed further in Table 1.

## Study design

The study was designed for participants to receive 4.5 mg/m$^2$ everolimus orally once daily for 7–21 days to attain plasma trough concentrations of 5–15 ng/ml, followed by surgical resection, and specimen collection of blood and brain tissue. After a minimum of three patients were treated for one week, if there was no evidence of serious adverse effects and other dose-level tolerability and safety issues, treatment duration would be extended by a maximum of two weeks. Due to most patients coming from long distances, everolimus treatment lasted an average of 7 days in the 4 Active participants. Ten Control participants did not receive everolimus for at least six months before surgery. Participants were monitored for adverse events, with evaluation of the planned surgical incision region inspected one week prior to everolimus treatment, 5 days after everolimus, and at weekly intervals from one to six weeks after surgery

for evidence of infection and abnormal wound healing by a neurosurgeon [18]. The surgical sites were photographed by a neurosurgeon at each incision evaluation and were then blindly reviewed for all participants to evaluate wound infection or dehiscence requiring surgery. Additionally, possible infection was monitored by blood work as indicated by abnormally elevated serum white blood cell count and C-reactive protein with no known cause, persistent fever $\geq 101°F$, or bacterial meningitis. Participants would be removed from the study with Grade 3 or 4 adverse events as defined by Common Terminology Criteria for Adverse Events (CTCAE) v.4.0, or if the investigator or patient decided continuation was not in the best interest of the patient. Changes to seizure frequency were not evaluated, given the brief study duration and small participant numbers with high intra-individual variability in seizure frequency.

## Specimen collection

After an average of 7 days everolimus (Active) or no intervention (Control), peripheral blood and brain tissue were collected at the time of surgical resection. Plasma was isolated by centrifugation from blood immediately and stored at -80°C. Plasma was used to quantify everolimus levels, proteomics, and cytokine analyses. A portion of the resected brain tissue was immediately frozen and stored at -80°C; another portion of the brain tissue was processed for formalin fixed paraffin embedding (FFPE) and reviewed for neuropathology diagnoses as summarized in Table 1.

## Everolimus detection

Plasma everolimus levels were quantified by ARUP Laboratories using the QMS Everolimus Immunoassay (ThermoScientific). The detection limit is < 2 ng/ml in plasma.

Brain tissue everolimus levels were evaluated by the NYU Metabolomics Core Resource Laboratory, with a lower detection threshold limit of 333 femtomoles everolimus per mg brain tissue by LC-MS. Everolimus drug metabolites (CYP450 products [19]) were not evaluated. Extraction buffer (90% ACN with 100 nM everolimus-d4) was added for homogenization to yield an extraction ratio of 10 mg brain tissue per 1 mL of extraction buffer. After homogenization, samples were spun down by table-top microcentrifuge and 450 μL of the supernatant was transferred to a new microcentrifuge tube and was then dried down to completion using SpeedVac. Ultimately, 50 μL of 100% ACN was used to resuspend the sample prior to LC-MS data acquisition. An everolimus standard curve with matrix control (blank mouse brain tissue) with everolimus concentration ranging from 1 μM to 3 nM (final concentration in homogenate) was processed side by side with surgical samples. In addition, blank mouse brain extractions with no everolimus spike in were also included as blank controls to determine the blank threshold level for the unlabeled everolimus. 100 nM of labeled everolimus (everolimus-d4) was also included in the extraction buffer as an internal control for all experimental, standard curve and blank sample extractions, and as a reference for instrument performance assessment during the data acquisition process. The scan range was from 900 m/z-1000 m/z with Full MS-SIM to maximize instrument time for each targeted everolimus ion (targeted everolimus ion: 908.5506 *m/z* and 975.6136 *m/z*; targeted everolimus-d4 ion: 912.5757 *m/z* and 979.6383 *m/z*. The resulting standard curve was linear ($R^2$ = 0.9990) and even the two curve points with the lowest everolimus spike in concentration (3.33 nM) have detected everolimus levels higher than the blank threshold. None of the surgical samples had an intensity for the unlabeled everolimus that was higher than the blank threshold cut off. The instrument performance for everolimus detection was stable throughout data acquisition period with a coefficient of variation of 12.6% across all samples.

## Western blot

Protein was isolated from frozen brain tissue (average 30 mg/sample) at 20% weight/volume in Tris-NaCl buffer (20 mM Tris base, 150 mM NaCl, 0.1% Triton-X 100, protease and phosphatase inhibitors at pH 7.5) using a hand held homogenizer equipped with a pestle on ice. Samples were incubated on ice for 15 minutes, centrifuged for 15 minutes at 14,000$g$, 4°C, and supernatant was isolated. Protein concentration was determined by BCA assay according to the manufacturer's protocol (Pierce). Lysates (25 μg/lane) were boiled in Bolt LDS Sample Buffer and DTT. One sample (NYU-C-001) was included on both gels to allow for normalization across blots for all samples. Proteins were resolved on a 4–12% Bis-Tris gel (Invitrogen) and transferred onto nitrocellulose membranes. After blocking in 5% milk TBST, blots were probed for phospho-S6 Ser235/236 (1:1000, Cell Signaling #2211), phospho-S6 Ser240/244 (1:1000, Cell Signaling #5364), total S6 (1:1000, Cell Signaling #2217), or actin (1:3000, Sigma A5441) in 5% milk TBST overnight at 4°C. Blots were incubated with corresponding HRP-conjugated secondary antibodies (1:3000, GE Healthcare) for 1 hour at room temperature. Bands were visualized after ECL (Pierce) on a BioRad ChemiDoc with the NYU Small Instrument Fleet. Blot images were analyzed in Fiji ImageJ for quantification. Statistical analysis (student's unpaired t-test) and graphs were generated in GraphPad Prism v.9.2.0.

## Immunohistochemistry

To evaluate ribosomal S6 protein expression in gray matter, immunohistochemistry was performed as described previously [20]. Briefly, FFPE blocks were sectioned (8 μm) by the NYU Center for Biospecimen Research and Development (CBRD). Sections were deparaffinized and rehydrated through a series of xylenes and ethanol dilutions, followed by heat-induced antigen retrieval with 10 mM sodium citrate and 0.05% Triton-X 100 at pH 6. Sections were blocked with 10% normal donkey serum with 0.05% Triton-X 100 and incubated with total S6 (1:100, Cell Signaling #2217), phospho-S6 Ser235/236 (1:100, Cell Signaling #2211), or phospho-S6 Ser240/244 (1:100, Cell Signaling #5364) primary antibodies with 0.05% Triton-X 100 overnight at 4°C. Corresponding secondary antibody was used (donkey anti-rabbit Alexa-Fluor 568, Thermofisher) with Hoescht counterstain in PBS for 2 hours at room temperature, and slides were coverslipped. Whole slide scanning was performed on each section at 20X magnification on a NanoZoomer HT2 (Hamamatsu) microscope with the NYU Experimental Pathology Research Laboratory. Images were analyzed at 5X magnification in Fiji ImageJ by the same binary threshold for all images to determine the number of positive pixels in each image, which was reported as an average percentage of the total image area (2–3 images per slide depending on tissue size). To calculate percent phospho-S6, the corresponding 5X region was evaluated in the parallel total S6 section. Statistical analysis (student's unpaired t-test) and graphs were generated in GraphPad Prism v.9.2.0.

## Brain proteomics

Proteomics was performed on approximately 10 mg frozen brain tissue in collaboration with the NYU Proteomics Laboratory.

**Tissue lysis and protein digestion.** Frozen brain tissue was lysed in the buffer composed of 5% SDS in 50 mM of TEAB at pH 7.5. Tissue lysis was assisted by sonication. Protein concentrations were measured with BCA assay. Samples were supplemented with 10 mM TCEP and 20 mM CAA and incubated at 95°C for 10 minutes. 50 μg of total protein from each sample were used for subsequent processing. Proteins were cleaned-up from SDS and digested into peptides following S-Trap protocol [21] according to the manufacturer recommendations (https://protifi.com/pages/s-trap). In brief, proteins were precipitated by acidification with

phosphoric acid (to final 1.2% (v/v)) and 6 times dilution with Strap buffer (90% methanol in 100 mM TEAB). Protein precipitate was retained on Strap spin columns and after washing with Strap buffer proteins were digested on-column with trypsin (Promega, sequencing grade, 1:50 w:w ratio) for 1 hour at 47°C. Resulting peptides were desalted on self-packed C18 Stage-Tips and dried in a SpeedVac.

**TMT peptide labeling.** For MS based quantification, we labeled peptides with TMT Pro tags [22] (ThermoScientific). Peptide samples were solubilized in 20 μl of 50 mM HEPES buffer (pH 8.5) and 8 μl of ACN stock of the TMTPro reagent (12 μg/μl conc.) was used for labeling. After 1 hour at room temperature, unreacted labels were quenched with 500 mM ammonium bicarbonate (ABC, 30 minutes at 37°C), samples were acidified to pH~2–3 with formic acid and pooled together. After dilution with water to lower ACN concentration to 2% salts were removed on C18 cartridges (tC18 SepPak, Waters) and peptide samples were concentrated in a SpeedVac and solubilized in 5% ACN 10 mM ammonium bicarbonate solution.

**Offline high-pH RP fractionation.** To increase depth of the proteome coverage TMT-labeled peptides were fractionated offline by high pH reverse phase chromatography [23] using a Waters XBridge BEH 130A C18 3.5um 4.63mm ID x 250 mm column on an Agilent 1260 Infinity series HPLC system. Peptides were separated by a linear gradient from 5% to 35% ACN in 62 minutes followed by a linear increase to 60% ACN in 5 minutes, and ramped to 70% ACN in 3 minutes. All buffers were contained 10 mM ABC in order to maintain high pH. Fractions were collected every 60 seconds. Fractions were combined in a nonconsecutive pooling scheme in order to get final 8 fractions. They were acidified with formic acid and concentrated using vacuum centrifugation.

**LC-MS/MS analysis.** Peptide fractions were analyzed by LC-MS/MS. LC separation was performed online on EASY-nLC 1000 (Thermo Scientific) utilizing Acclaim PepMap 100 (75 μm x 2 cm) precolumn and PepMap RSLC C18 (2 μm, 100A x 50 cm) analytical column. Peptides were gradient eluted from the column directly to Orbitrap Q-Exactive mass spectrometer at a flowrate of 200 nl/min. High resolution full MS spectra were acquired with a resolution of 120,000, an AGC target of 3e6, with a maximum ion injection time of 100 ms, and scan range of 400 to 1600 m/z. Following each full MS scan 20 data-dependent HCD MS/MS scans were acquired at the resolution of 60,000, AGC target of 5e5, maximum ion time of 100 ms, one microscan, 0.4 m/z isolation window, Normalized Collision Energy (NCE) of 30, fixed first mass at 100 m/z and dynamic exclusion for 45 seconds.

**Data analysis.** MS data were analyzed using MaxQuant software version 1.6.3.4 [24] and searched against the SwissProt subset of the human Uniprot database (http://www.uniprot.org/) containing 20,430 entries. Database search was performed in Andromeda [25] integrated in MaxQuant environment. A list of 248 common laboratory contaminants included in MaxQuant was also added to the database as well as reversed versions of all sequences. For searching, the enzyme specificity was set to trypsin with the maximum number of missed cleavages set to 2. The precursor mass tolerance was set to 20 ppm for the first search used for non-linear mass re-calibration [26] and then to 6 ppm for the main search. Oxidation of methionine was searched as variable modification; carbamidomethylation of cysteines was searched as a fixed modification. TMT labeling was set to lysine residues and N-terminal amino groups, corresponding batch-specific isotopic correction factors were accounted for. The false discovery rate (FDR) for peptide, protein, and site identification was set to 1%, the minimum peptide length was set to 6. For statistical analysis of the TMT data MSstatsTMT R package was used [27]. The mass spectrometry raw files are accessible under MassIVE ID: (MSV000088551).

Subsequent data analysis was performed in either Perseus [28] (http://www.perseus-framework.org/) or using R environment for statistical computing and graphics (http://www.r-project.org/) and detailed summary in S1 and S2 Tables.

## WGCNA

WGCNA was performed to determine whether brain proteomics correlated to phospho-S6 levels (as quantified in whole brain homogenate on western blot) in the R environment with the *WGCNA* package for blockwiseModules with defaults except where noted, similar to previously described [29, 30]. Soft threshold power beta was determined at $R^2 = 0.8$ (power = 11). Gene ontology annotations for modules was determined following WGCNA with the *anRichment* package in the R environment with Entrez IDs against the human GOcollection (S3 Table). GO annotations were considered with an FDR < 5% and associated with at least 5 proteins.

## Brain metabolomics

Global polar LC-MS was performed on the same brain tissue sample for everolimus quantification by the NYU Metabolomics Core Resource Laboratory, detailed in S4 Table. Briefly, brain homogenate from everolimus analysis were extracted with 80% ACN and subjected to LC-MS global polar analysis. Untargeted feature-based analysis was performed to expand potential scope of discovery. Representative LC-MS features were detected (PPM = 20.0, RT_DELTA = 2.0, MIN_SIGNAL = 100000, MIN_RANGE = 10.0), and resulting list was used for relative peak quantification. Overall, 849 features were quantified across all 14 samples. In order to identify putative molecules of interest, the available MS/MS spectra were searched against a data analysis pipeline including both the NIST17MS/MS [31] and METLIN [32] spectral library databases and high-scoring hits (RevDot > 900) were retained. Metabolites were reported as detected if their intensity was above the blank threshold (determined by blank replicates, 3X Signal/Noise), and metabolites that were not above this detection limit in all samples were discarded. In addition, duplicate metabolite names that were detected at different retention times in a single sample were retained with an RT identification. For annotation of metabolite features without a putative spectral library match, we used a variety of manual approaches including spectral library search, accurate mass formula search, and isotope fine structure analysis.

## Plasma proteomics

**Immunodepletion and enzymatic digestion.** Plasma samples (8 μl, ~ 400 μg of protein) were immunodepleted from high abundant proteins using High Select Top14 Abundant Protein Depletion Mini Spin Columns (ThermoScientific) according to the manufacturer protocol. After a single step immunodepletion, plasma samples were supplemented with sodium deoxycholate (SDC, to final 1%), 20 mM 2-chloroacetamide and 10 mM tris (2-carboxyethyl) phosphine and incubated for 30 minutes at 95°C with subsequent digestion by trypsin (Promega Sequencing Grade, 1:50 w:w ratio, overnight at 37°C). SDC was pelleted upon samples acidification (with FA, to final 1%) and removed. Peptides were desalted on C18 SepPak cartridges (Waters) and dried on SpeedVac.

**LC-MS/MS analysis.** LC separation was performed online on EASY-nLC 1000 (Thermo Scientific) utilizing Acclaim PepMap 100 (75 μm x 2 cm) precolumn and PepMap RSLC C18 (2 μm, 100A x 50 cm) analytical column. Peptides were gradient eluted from the column directly to Orbitrap HFX mass spectrometer. Flowrate was set to 200 nl/min. The mass spectrometer was operated in data-independent acquisition mode (DIA) [33] doing MS$^2$ fragmentation across 22 m/z windows after every MS$^1$ scan event (S5 Table).

High resolution full MS spectra were acquired with a resolution of 120,000, an AGC target of 3e6, with a maximum ion injection time of 60 ms, and scan range of 350 to 1650 m/z.

Following each full MS scan, data-independent HCD MS/MS scans were acquired at the resolution of 30,000, AGC target of 3e6, stepped NCE of 22.5, 25 and 27.5.

**Data analysis.** MS data were analyzed using Spectronaut software (https://biognosys.com/shop/spectronaut) and searched in directDIA mode against the SwissProt subset of the human Uniprot database (http://www.uniprot.org/). Database search was performed in integrated search engine Pulsar. For searching, the enzyme specificity was set to trypsin with the maximum number of missed cleavages set to 2. Oxidation of methionine was searched as variable modification; carbamidomethylation of cysteines was searched as a fixed modification. The FDR for peptide, protein, and site identification was set to 1%. Protein quantification was performed on MS2 level using 3 most intense fragment ions per precursor. BGS default search/quantification scheme was used. The mass spectrometry raw files are accessible under MassIVE ID: (MSV000088551).

Subsequent data analysis was performed in either Perseus [28] (http://www.perseus-framework.org/) or using the R environment for statistical computing and graphics (http://www.r-project.org/) and detailed summary in S6 and S7 Tables.

## Plasma cytokines

Total VEGF was evaluated in plasma by ARUP Laboratories with a Quantitative Chemiluminescent Immunoassay. VEGF-D (plasma 1:3 dilution in assay buffer; Millipore HAGP1MAG-12k-01), IL-18 (plasma neat; Millipore HCYTA-60K-07), and high-sensitivity 6-plex [plasma neat; IL-1 beta, IL-6, IL-8, IL-12 (p70), IFN-gamma, TNF-alpha; Millipore HSTCMAG-28SK] were evaluated by the manufacturer's protocol, including validation by internal quality controls. Cytokine levels were measured with the Luminex 200 and data was analyzed by Belysa Immunoassay Curve Fitting with the NYU Precision Immunology core. Statistical analysis (student's unpaired t-test) and graphs were generated in GraphPad Prism v.9.2.0.

## Results

### Case history

Participant case history is summarized in Table 1. Plasma everolimus levels at the time of surgical resection were on average 12.4 ng/ml, and within the targeted range of 5–15 ng/ml (Table 1). Brain everolimus was below the detectable limit of 333 femtomoles everolimus per mg brain. No adverse or serious adverse events were reported in any participants. In general, all participants tolerated everolimus with minimal side effects reported.

### Ribosomal S6 protein phosphorylation

To evaluate whether phospho-S6 was lower in surgically resected brain tissue of Active participants compared to Control participants, we evaluated percent phospho-S6 of total S6 in whole brain homogenate and histologically.

Western blot of whole brain homogenate indicated variable levels of total S6, particularly among the Active group ($p = 0.97$, Fig 1A and 1B). Further, there is a smaller prominent second band detected for total S6 in NYU-002. The percent phospho-S6 relative to total S6 trended toward lower values in the Active versus Control participants; our study was not powered to detect small differences. The percent phospho-S6 (Ser235/236) relative to total S6 was on average 1.19-fold lower when comparing Active to Control participants ($p = 0.67$, Fig 1A and 1C). The percent phospho-S6 (Ser240/244) relative to total S6 was on average 1.15-fold lower when comparing Active to Control participants ($p = 0.66$, Fig 1A and 1D).

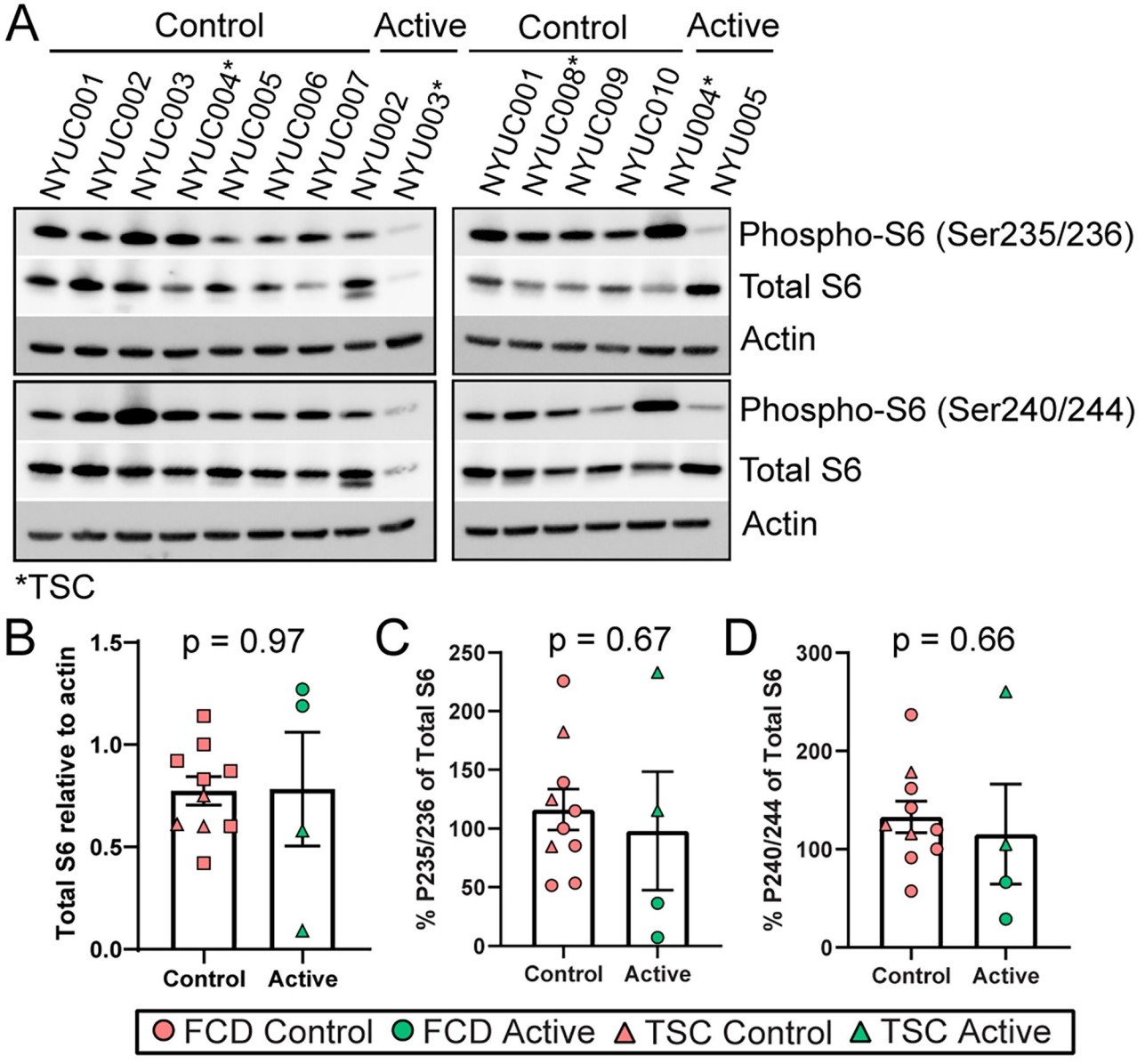

**Fig 1. Ribosomal S6 phosphorylation in whole brain homogenate. A)** Representative western blot images from surgically resected brain tissue of Active participants after taking 4.5 mg/m$^2$ everolimus for 7 days and controls. Total S6, phospho-S6 (Ser235/236), phospho-S6 (Ser240/244), and beta-actin as loading control were evaluated in 14 participants. TSC participants are indicated "∗", while all others are FCD participants. **B)** Quantification of total S6 relative to actin indicates variability in baseline levels from sampled brain tissue, as well as one patient with increased expression of a smaller detected band (NYU-002). **C)** Percentage of phospho-S6 (Ser235/236) relative to total S6 indicates an average 1.19-fold decrease when comparing all Active participants to controls (p = 0.67). The highest level was seen in NYU-004. **D)** Percentage of phospho-S6 (Ser240/244) relative to total S6 indicates an average 1.15-fold decrease when comparing all Active participants to controls (p = 0.66). The highest level was seen in NYU-004. Error bars indicate SEM.

Histologically, there were also variable levels of baseline total S6 in gray matter (p = 0.90, Fig 2). The percent phospho-S6 (Ser235/236) relative to total S6 in parallel sections was on average 1.56-fold lower when comparing Active to Control participants (p = 0.37, Fig 2N). The percent phospho-S6 (Ser240/244) relative to total S6 was on average 5.55-fold lower when comparing Active to Control participants (p = 0.22, Fig 2O).

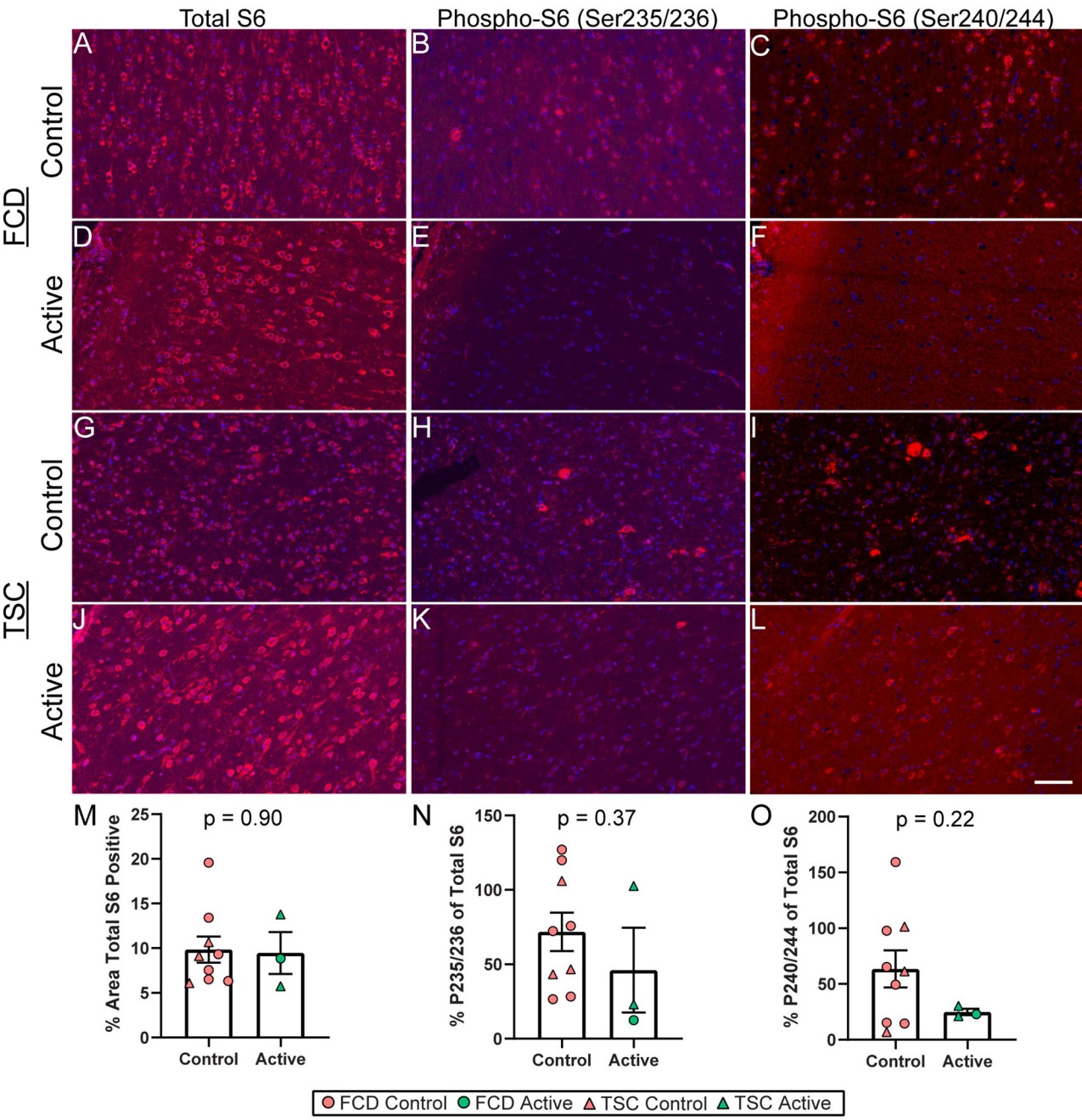

**Fig 2. Histological quantification of ribosomal S6 phosphorylation in gray matter.** Representative images show total S6, phospho-S6 (Ser235/236), and phospho-S6 (Ser240/244) (red) in resected cortical tissue from FCD participants that were Controls **A-C)** or Active **D-F)** and in TSC participants that were Controls **G-I)** or Active **J-L). M)** Quantification of total S6 expression in the gray matter indicated no difference in baseline expression between Active and Control participants. **N)** Quantification of percent phospho-S6 (Ser235/236) relative to total S6 in the gray matter indicated an average 1.56-fold decrease when comparing all Active participants to Controls (p = 0.37). **O)** Quantification of percent phospho-S6 (Ser240/244) relative to total S6 in the gray matter indicated an average 5.55-fold decrease when comparing all Active participants to Controls (p = 0.22). Histology was performed on all cases with available FFPE, excluding 1 Active FCD case. Error bars indicate SEM. Scale bar represents 100 μm.

## Brain proteomics

In surgically resected brain tissue, there were 2470 proteins detected by the TMT method (S1 Table). PCA indicated no segregation of Active from Control participants in PCA1 (p = 0.73)

and some segregation in PCA2 (p = 0.016, Fig 3A). There was significant enrichment for 357 signaling pathways (p value of overlap < 0.05) associated with the 2470 proteins (S2 Table). The most enriched pathway was EIF2 signaling (p value of overlap = 3.16 x $10^{-51}$, z = -0.77). There was enrichment for mTOR signaling related pathways and a trend in inhibition as determined by the z score, including EIF2 signaling, regulation of eIF4 and p70S6K signaling (p value of overlap = 1.26 x $10^{-32}$, z = -2.04), mTOR signaling (p value of overlap = 1.58 x $10^{-28}$, z = -2.72), and p70S6K signaling (p value of overlap = 1.00 x $10^{-14}$, z = -2.72). There were no proteins differentially expressed when comparing Active and Control participants at an FDR<5%; thus, there were no significantly altered signaling pathways (|z| > 2) associated with the proteins detected (mTOR pathway proteins RPS6KB1, PTEN, TSC1, TSC2 were not detectable; Fig 3B). FKBP1A, MTOR, and RPS6 were detected with no differences after everolimus. There were 11 proteins differentially expressed at FDR<15%, with 4 proteins (FGA, FGB, FGG, PLG) associated with activation of coagulation system (p value of overlap = 1.45 x $10^{-9}$, z = 2.00) and acute phase response signaling (p value of overlap = 1.23 x $10^{-6}$, z = 2.00; Fig 3B, S1 Fig). The histology quantification of total S6 (RPS6, p = 0.90, Fig 2) provided validation of the proteomics results (p = 0.94, S1 Table) with some variability in the Active group that is also seen by western blot (p = 0.97, Fig 1B).

WGCNA was performed to evaluate whether increasing phospho-S6 was associated with altered proteins in mTOR signaling related signaling pathways, i.e. translation, or other pathways regardless of everolimus and clinical diagnoses. WGCNA indicated that there were 11 modules associated with the 2470 proteins. There were 1109 proteins that correlated with phospho-S6 (P235/236, p < 0.05), distributed across 10 modules (all but M-Purple). The top significantly correlated protein with P235/236 levels was a positive correlation to HSPA2 (p = 1.18 x $10^{-6}$, $R^2$ = 0.87; Fig 3C) in the M-Brown module. Phospho-S6 (P240/244) also correlated with HSPA2, but to a lesser extent (p = 0.0052, $R^2$ = 0.49). There were 625 proteins that correlated with P240/244 (p < 0.05), distributed across 10 modules (all but M-Pink). The top significantly correlated protein with P240/244 was a negative correlation with ANK2 (p = 7.92 x $10^{-5}$, $R^2$ = 0.74; Fig 3D) in the M-Brown module. P235/236 also correlated with ANK2, but to a lesser extent (p = 0.0014, $R^2$ = 0.59). Module trait analysis following WGCNA indicated that 5 modules significantly correlated (p < 0.05) to phospho-S6 (Fig 3E). There were 2 modules (M-Green, M-Brown) with a negative correlation and 3 modules (M-Yellow, M-Turquoise, M-Black) with a positive correlation to increasing phospho-S6. M-Green had a negative correlation with P235/236 (p = 3.93 x $10^{-4}$, corr. = -0.81) and P240/244 (p = 3.84 x $10^{-2}$, corr. = -0.56), with top GO annotation of cellular respiration (S3 Table). M-Brown had a negative correlation with P235/236 (p = 1.04 x $10^{-3}$, corr. = -0.78) and P240/244 (p = 1.94 x $10^{-2}$, corr. = -0.61), with top GO annotation of synaptic transmission. M-Yellow had a positive correlation with P235/236 (p = 1.78 x $10^{-3}$, corr. = 0.76), with top GO annotation of organophosphate metabolic process. M-Turquoise had a positive correlation with P235/236 (p = 1.03 x $10^{-2}$, corr. = 0.66) and P240/244 (p = 1.91 x $10^{-2}$, corr. = 0.62), with top GO annotation of nuclear mRNA catabolic process. M-Black had a positive correlation with P235/236 (p = 2.94 x $10^{-2}$, corr. = 0.58), with top GO annotation of neuron ensheathment.

## Brain metabolomics

Global polar metabolomics analysis by LC-MS was performed in frozen surgically resected brain tissue. PCA indicated no segregation of Active and Control participants in PCA1 (p = 0.32) or PCA2 (p = 0.48; Fig 4A). There were 14 metabolite features increased (p<0.05, $\log_2$(fold change) > 1) in Active participants (Fig 4B and 4C, S4 Table), with the most frequent annotations for N-acetylaspartylglutamic acid (NAAG).

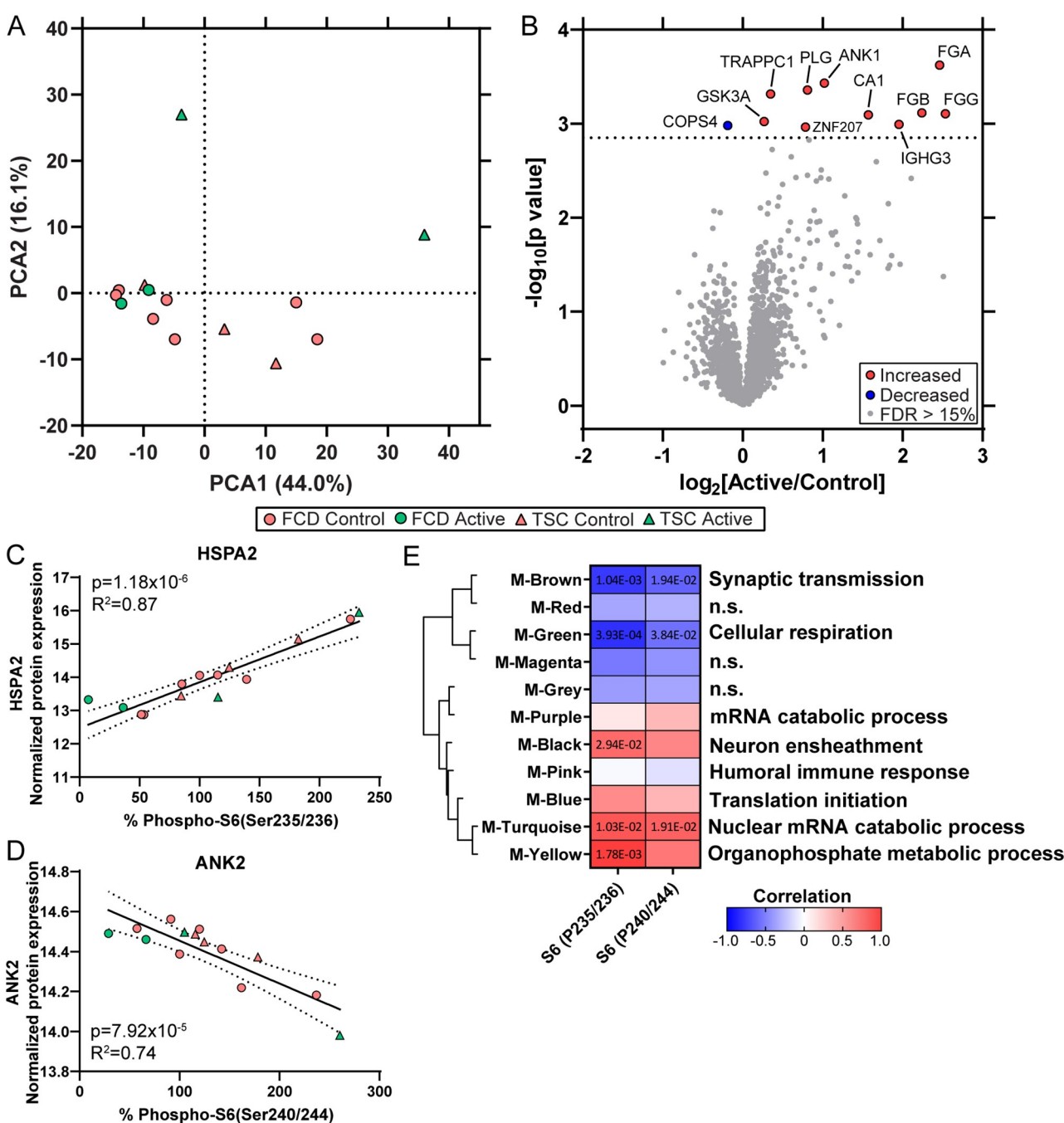

**Fig 3. Proteomics in whole brain homogenate.** Proteomics was performed in surgically resected brain tissue from Active and Control participants. **A)** PCA in brain tissue indicated no segregation of Active and Control participants in PCA1 (p = 0.73) and with some segregation in PCA2 (p = 0.016). Clinical diagnoses (FCD, TSC) are noted as well. **B)** Differential expression analysis in brain tissue indicated that there were 11 significantly altered proteins in Active versus Control participants at an FDR < 15% (dotted line), detailed in S1 Table and S1 Fig. There were no significant proteins at FDR < 5–10%. **C)** WGCNA of brain proteomics and phospho-S6 evaluated in total brain homogenate by western blot, regardless of everolimus and clinical diagnoses. There were 1109 proteins that correlated with phospho-S6 levels (P235/236, p < 0.05), distributed across 10 modules (all but M-Purple). The top significantly correlated protein with P235/236 was a positive correlation to HSPA2 (p = 1.18 x $10^{-6}$, $R^2$ = 0.87) in the M-Brown module. **D)** The top significantly correlated protein with P240/244 was a negative correlation with ANK2 (p = 7.92 x $10^{-5}$, $R^2$ = 0.74) in the M-Brown module. **E)** Module trait correlation identified 5 significantly associated modules with phospho-S6 (p < 0.05). Modules were clustered by eigenprotein adjacency (relatedness to other modules) on the left. Name of module is indicated by "M-color." P values are indicated for those modules with p < 0.05 correlation. Positive correlation is indicated in red and negative correlation in blue. Top module GO annotations are noted on the right (FDR < 5%, at least 5 proteins/annotation). Several modules did not have a significant GO annotation ("n.s.").

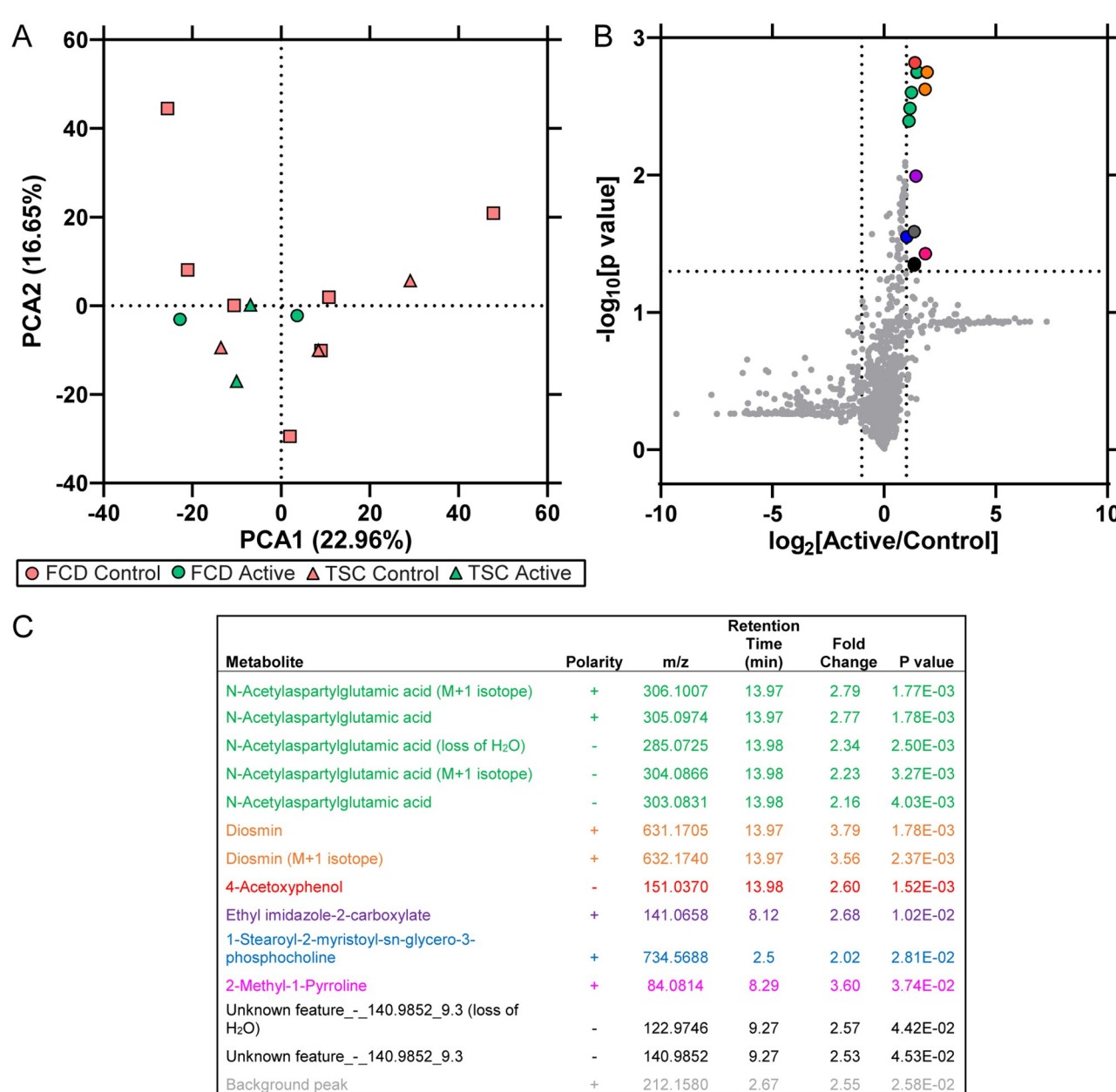

**Fig 4. Metabolomics in whole brain homogenate.** Global polar metabolomics by LC-MS was performed on surgically resected brain tissue from Active and Control participants. **A)** PCA indicated distribution of participants, as well as clinical diagnoses (FCD, TSC). There was no segregation of Active and Control participants in PCA1 (p = 0.32) or PCA2 (p = 0.48). **B)** Differential expression analysis (p < 0.05, $\log_2$ (fold change) > 1) identified 14 metabolite features that were increased in Active participants, associated with the indicated annotations detailed in **C)**.

## Plasma proteomics

Proteomics on plasma obtained at the time of surgical resection identified 671 proteins by LFQ DIA method (S6 Table). PCA indicated no segregation of Active from Control participants (Fig 5A) in PCA1 (p = 0.67) or PCA2 (p = 0.37). There was significant enrichment for 164 signaling pathways (p value of overlap < 0.05) associated with the 671 proteins (S7 Table). The most enriched pathway was LXR/RXR activation (p value of overlap = 2.51 x $10^{-49}$,

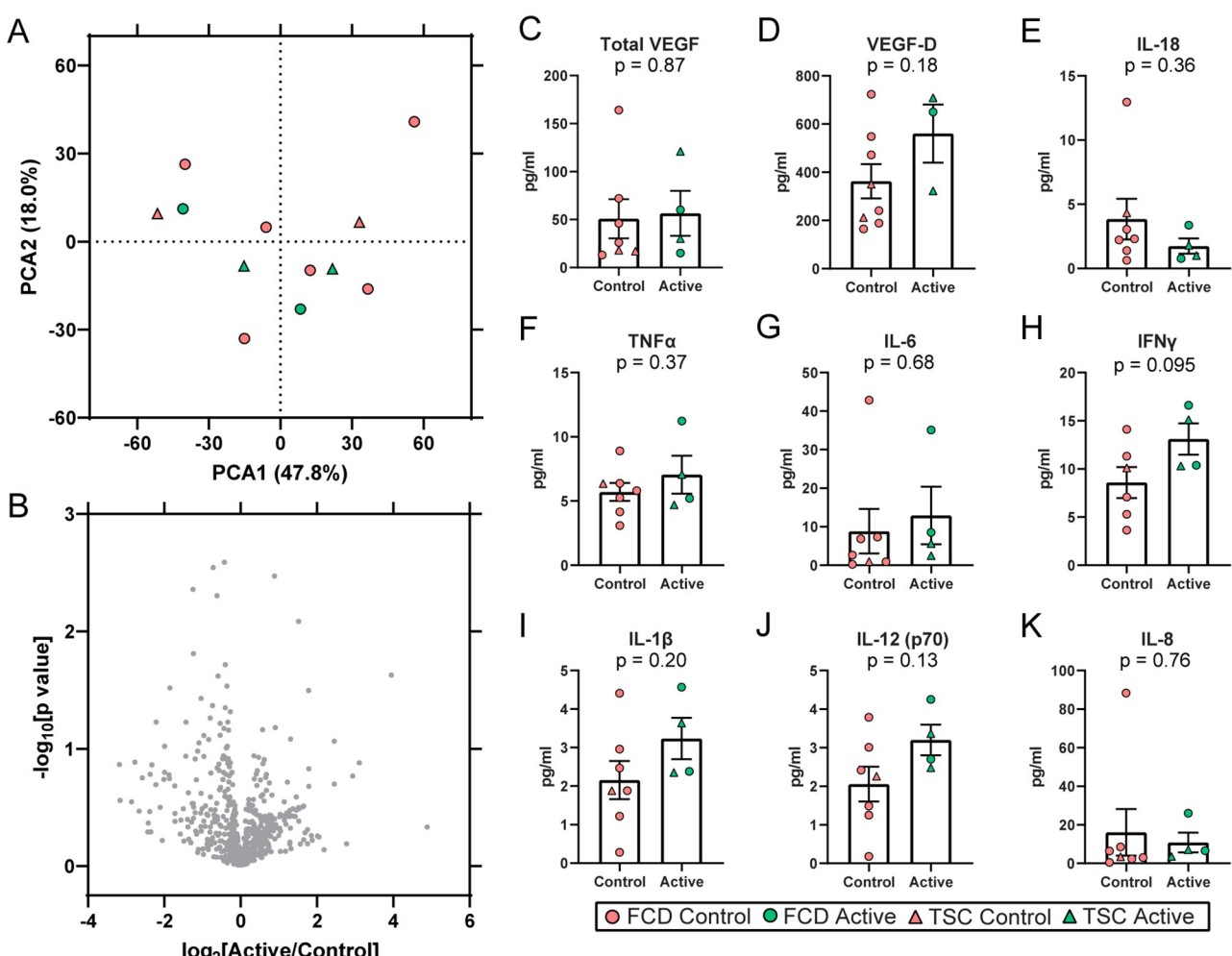

**Fig 5. Plasma proteomics and cytokine levels. A)** Proteomics was performed on plasma obtained at time of surgical resection after 7 days everolimus. PCA indicates no segregation of Active from Control participants, nor by clinical diagnoses (FCD, TSC). **B)** Differential expression analysis of plasma proteins indicated no altered proteins in Active versus Control participants at FDR < 5%, nor at FDR < 15%. **C-K)** Plasma cytokine levels were evaluated at time of surgical resection. There were no significant differences in the cytokines evaluated.

z = -2.22). There was enrichment for mTOR signaling related pathways, including EIF2 signaling (p value of overlap = $2.14 \times 10^{-8}$, z = -1.26), regulation of eIF4 and p70S6K signaling (p value of overlap = $1.23 \times 10^{-5}$, z = N/A), p70S6K signaling (p value of overlap = $8.71 \times 10^{-5}$, z = 0.00), and mTOR signaling (p value of overlap = $2.69 \times 10^{-3}$, z = N/A). Differential expression analysis at an FDR<5% did not identify significantly altered proteins (nor at FDR<15%), thus there were no significantly altered signaling pathways ($|z| > 2$) as a result of everolimus in plasma associated with the proteins detected (Fig 5B, S6 Table).

## Plasma cytokines

Plasma cytokines were analyzed from samples obtained at the time of surgical resection, with no differences between Active and Control participants (Fig 5C–5K).

## Discussion

This pilot study of everolimus in TSC and FCD patients with TRE undergoing surgical resection revealed no adverse events and detectable plasma everolimus in all Active participants. There was a trend of lower brain phospho-S6 in Active compared to Control participants. The broader systemic effect of everolimus indicated no significant proteins at 5% FDR by brain proteomics, with trends in reduced mTOR signaling related pathways. Among all participants, higher phospho-S6 was associated with decreased synaptic transmission and cellular respiration as well as increased neuron ensheathment, nuclear mRNA catabolic process, and organophosphate metabolic process. Brain metabolomics identified increased metabolites associated with modified glutamate receptor signaling. There were no differences between Active and Control participants in plasma proteomics, VEGF, or other targeted cytokines.

Previous and ongoing everolimus trials in TSC and FCD excluded surgical cases [14, 15, 34]. EXIST clinical trials in TSC indicated safety, decreased SEGA volume (EXIST-1 and EXIST-2, 4 year studies), and reduced seizures (EXIST-3, 1.5 year study) at similar dosing to our study [14, 15]. We similarly found no safety concerns, with no reported adverse events before, during, or after surgical resection. Impaired wound healing has been reported in some patients receiving everolimus or other mTOR inhibitors [35], particularly for patients with higher body mass index (BMI). We specifically examined scalp wound healing in study participants and found no evidence for presence of this complication. One additional active study in Korea evaluated everolimus in FCD type II but excluded surgical cases, with no results yet (NCT03198949). Finally, a study evaluating the analog rapamycin (2–10 mg daily for one week) in PTEN-deficient glioblastoma patients undergoing surgical resection reported no perioperative bleeding complications [16].

We identified a trending decrease in phospho-S6 after everolimus in both whole brain homogenate and histologically. Levels varied between detection techniques likely due to tissue availability, subregional differences, and inherent detection method specificity (i.e. solubility). Similarities in brain phospho-S6 between western blot and histology was reported after rapamycin in glioblastoma, with substantial variation in intra-tumoral phospho-S6 relative to baseline surgery but with agreement between methods and differing expression magnitudes with more dramatic differences by western blot [16]. It has also been suggested that total S6 may be influenced by mTOR dependent translation, in which total S6 was reduced in 2/3 patients after rapamycin [16]. Phosphorylation of Ser235/236 and Ser240/244 may differ as well. Phospho-S6 (Ser235/236) is phosphorylated by both mTOR dependent (via S6 kinase) and independent (via p90 ribosomal S6 kinases) mechanisms [36]. There may also be individual differences, as seen in the rapamycin study (7/14 patients with inhibition of tumor cell proliferation) where these differences were found to be host-related (cell type/tissue specific sequestration) rather than cell intrinsic (mutation of drug target), as seen by successful rapamycin induced mTOR inhibition of *ex vivo* tumor cultures [16].

We identified proteins that correlated to increased phospho-S6 by WGCNA. M-brown was associated with decreased synaptic transmission with proteins involved in transport and localization, including the most significantly correlated proteins HSPA2 and ANK2. HSPA2 is a part of the HSP70 family that removes damaged or dysfunctional proteins, particularly during cellular stress response and neurodegeneration [37, 38]. In epilepsy patients and animal models, induced hippocampal HSP70 was related to stress response rather than a protective response [39, 40], and HSP70-positive neurons were increased in sudden unexpected death in epilepsy (SUDEP) when compared to non-epilepsy [41]. HSP70 upregulation also resulted in proteasomal degradation of voltage gated $K_v4$ potassium channels (KCND2, KCND3), and HSP70 knockdown in an animal model suppressed seizures and prevented $K_v4$ potassium

channel degradation [40]. ANK2 is a cytoskeleton scaffolding protein that coordinates protein assembly, drives axonal branching, influences connectivity in neurons [42], and is known to play a role in long QT syndrome as a result of abnormal ion channel localization [43, 44]. Previous studies indicate that mTOR pathway mutations elevated phospho-S6, impaired autophagy, disrupted neuronal ciliogenesis, and abrogated Wnt signaling necessary for neuronal polarization [45]. The mTOR signaling pathway is further implicated in synaptogenesis in normal development and in an epilepsy model, in which increased phospho-S6 levels were associated with dendritic damage that could be reversed by rapamycin [46, 47].

In addition to decreased synaptic transmission by WGCNA, we also identified that increased phospho-S6 correlated to decreased cellular respiration as well as increased nuclear mRNA catabolic process and organophosphate metabolic process. M-green, the most significantly correlated module to P235/236, was associated with decreased mitochondrial proteins involved in cellular respiration. Previous studies indicate mTOR promotes translation of nuclear encoded mitochondrial proteins to increase ATP levels [48]. We did identify a positive correlation of organophosphate metabolic processes (M-yellow) and increased phospho-S6, which may be related to increased synthesis of cofactors needed for DNA, RNA, or ATP molecules. Further, M-turquoise was associated with many ribosomal proteins, most of which were annotated as part of nuclear mRNA catabolic processes. This is a process in which nonsense mediated decay for mRNA with a nonsense codon prevents further translation [49]. Overall, there were increased ribosomal proteins, as might be expected after increased phospho-S6, but there may also be some regulation of abnormal mRNAs as has been reported in association with the mTOR pathway [50]. These protein changes should be investigated further in additional cases to determine the role of clinical factors, i.e. negative feedback loops particularly in the more benign tumor development in TSC [51] as well as other genetic risk factors.

Further, WGCNA correlated increased myelination proteins to higher phospho-S6 (M-black). Increased mTOR signaling does positively regulate myelination, through both protein and lipid synthesis [52]. Hypomyelination has been observed in models with knockdown of mTOR pathway genes (i.e. Raptor, Rictor, mTOR). The mTOR signaling pathway is also required in immature oligodendrocyte differentiation, as demonstrated *in vitro* by gene knockdown and after rapamycin. In the current study, whole brain homogenate analyses (both phospho-S6 by western blot and proteomics) may contain both neuronal and oligodendroglial cell types.

Prior everolimus studies found reduced polyamine metabolites needed for DNA packaging in cell proliferation and inhibited glycolytic activity in a cancer animal model [53]. *In vitro*, rapamycin altered metabolites that reflected a decrease in mitochondrial metabolism and a shift toward anaerobic glycolysis and lipid synthesis [54]. We identified increased metabolites, with most frequent annotations for NAAG. NAAG is a neurotransmitter co-released with others, like glutamate, and activates the metabotropic glutamatergic receptor 3 (GRM3) expressed by neurons and glia [55]. Post-synaptic neuronal GRM3 activation by NAAG results in inhibition of downstream signaling, resulting in reduced release of glutamate by post-synaptic neurons and thus may modify neuronal activity in conjunction with co-released glutamate. In epilepsy models, inhibition of the NAAG degradation enzyme (glutamate carboxypeptidase II) and GRM3 agonists module seizure frequency [56], and activation of GRM3 in traumatic brain injury models indicates a protective effect. With expression in astrocytes, it has been suggested that NAAG may also play a role in mediating inflammatory response.

Plasma cytokines in TSC patients with daily 5 or 10 mg everolimus for 48 weeks reflected decreased VEGF-D, decreased collagen type IV, and increased VEGF-A after 24 weeks that was maintained at lower levels through treatment [57]. *In vitro*, VEGF and IL-8 were decreased from neutrophils in healthy donors with increasing everolimus (1, 10, 100 ng/ml) with and

without TNFα induced inflammation [58], and IL-6 was increased from human macrophages after 10 μM everolimus for 24 hours [59]. In our short-term study, we did not identify cytokine differences, with some variability among participants.

There were several limitations to this study. There was a low participant number due to recruitment difficulties (few eligible candidates, few parents considered participation, many traveling long distances), several heterogeneous variables (age, brain region, neuropathology), no baseline blood or brain tissue analyses, short-term treatment, inclusion of FCD patients that may not have mTORopathies, and lack of genetic risk evaluation.

In summary, Active participants did not experience adverse events, targeted 5–15 ng/ml plasma everolimus was achieved, there was a trend in decreased brain mTOR signaling, and elevated brain phospho-S6 correlated to protein clusters associated with shifted energy production and myelination. Future studies should evaluate adverse events with long-term everolimus in patients undergoing surgical resection, additional cases (with neuropathological confirmation of FCD type I or II), and whether decreased brain phospho-S6 is associated with decreased seizure frequency.

## Supporting information

**S1 Table. LC-MS/MS proteomics in brain.**
(XLSX)

**S2 Table. LC-MS/MS brain IPA pathways.**
(XLSX)

**S3 Table. WGCNA GOanRichment in brain.**
(XLSX)

**S4 Table. Metabolomics raw data in brain.**
(XLSX)

**S5 Table. LC-MS/MS plasma sample DIA settings.**
(XLSX)

**S6 Table. LC-MS/MS proteomics in plasma.**
(XLSX)

**S7 Table. LC-MS/MS plasma IPA pathways.**
(XLSX)

**S1 Fig. Altered brain proteins at FDR<15% in Active versus Control participants. A-K)** Expression of the 11 altered brain proteins at FDR<15% in Active versus Control participants, in order by increasing p value. Distribution indicates that these differences are not all due to Active TSC participants, in which there was segregation of TSC cases on the PCA in Fig 3A. Error bars indicate SEM. **L)** The 11 altered proteins were associated with 2 signaling pathways having a p value of overlap < 0.05 and |z| > 2. The same four proteins (FGA, FGB, FGG, PLG) were associated with activation of the coagulation system (p value of overlap = 1.45 x $10^{-9}$, z = 2.00) and acute phase response signaling (p value of overlap = 1.23 x $10^{-6}$, z = 2.00). (TIF)

**S1 Raw image. Whole western blot images for phospho-S6 (Ser235/236). A)** All cases (n = 14) were evaluated by western blot for phospho-S6 (Ser235/236) quantification on 2 blots, as depicted in Fig 1A. Bands were visualized after ECL on a BioRad ChemiDoc. Quantification was performed on bands in the red outlined box in Fiji ImageJ. One sample (NYUC001) was included on both gels to allow for normalization across blots for all samples. **A')**

Corresponding ladder for panel A is shown in the colorometric brightfield image. **B)** On the same blots, actin was evaluated after stripping the phospho-S6 (Ser235/236). **B')** Corresponding ladder for panel B is shown in the colorometric brightfield image. **C)** On the same blots, total S6 was evaluated with no stripping (both actin and total S6 are present). **C')** Corresponding ladder for panel C is shown in the colorometric brightfield image.
(PDF)

**S2 Raw image. Whole western blot images for phospho-S6 (Ser240/244). A)** All cases (n = 14) were evaluated by western blot for phospho-S6 (Ser240/244) quantification on 2 blots, as depicted in Fig 1A. Bands were visualized after ECL on a BioRad ChemiDoc. Quantification was performed on bands in the red outlined box in Fiji ImageJ. One sample (NYUC001) was included on both gels to allow for normalization across blots for all samples. **A')** Corresponding ladder for panel A is shown in the colorometric brightfield image. **B)** On the same blots, actin was evaluated after stripping the phospho-S6 (Ser240/244). **B')** Corresponding ladder for panel B is shown in the colorometric brightfield image. **C)** On the same blots, total S6 was evaluated with no stripping (both actin and total S6 are present). **C')** Corresponding ladder for panel C is shown in the colorometric brightfield image.
(PDF)

## Author Contributions

**Conceptualization:** Orrin Devinsky.

**Data curation:** Dominique F. Leitner, Evgeny Kanshin, Manor Askenazi, Yik Siu, Drew Jones, Beatrix Ueberheide.

**Formal analysis:** Dominique F. Leitner, Evgeny Kanshin, Manor Askenazi, Yik Siu, Drew Jones, Beatrix Ueberheide.

**Funding acquisition:** Orrin Devinsky.

**Investigation:** Daniel Friedman, Orrin Devinsky.

**Methodology:** Dominique F. Leitner, Evgeny Kanshin, Manor Askenazi, Yik Siu, Drew Jones, Beatrix Ueberheide, Orrin Devinsky.

**Project administration:** Sasha Devore.

**Resources:** Thomas Wisniewski.

**Supervision:** Orrin Devinsky.

**Validation:** Dominique F. Leitner.

**Visualization:** Dominique F. Leitner.

**Writing – original draft:** Dominique F. Leitner.

**Writing – review & editing:** Dominique F. Leitner, Evgeny Kanshin, Manor Askenazi, Yik Siu, Daniel Friedman, Sasha Devore, Drew Jones, Beatrix Ueberheide, Thomas Wisniewski, Orrin Devinsky.

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
