## [Decision Letter · Decision Letter 0]

7 Mar 2022

PONE-D-22-02043Pilot Study Evaluating Everolimus Molecular Mechanisms in Tuberous Sclerosis Complex and Focal Cortical DysplasiaPLOS ONE

Dear Dr. Devinsky,

Thank you for submitting your manuscript to PLOS ONE. After careful consideration, we feel that it has merit but does not fully meet PLOS ONE’s publication criteria as it currently stands. Therefore, we invite you to submit a revised version of the manuscript that addresses the points raised during the review process.

 Please address all Reviewers' comments and make clear in the manuscript the limitations of the study

We look forward to receiving your revised manuscript.

Kind regards,

Emilio Russo

Academic Editor

PLOS ONE

Journal Requirements:

[I have read the journal's policy and the authors of this manuscript have the following competing interests:  All the authors report no conflicts of interest. 

Below are the disclosures of the three authors: 

Daniel Friedman receives salary support for consulting and clinical trial related activities performed on behalf of The Epilepsy Study Consortium, a non-profit organization. Dr. Friedman receives no personal income for these activities. NYU receives a fixed amount from the Epilepsy Study Consortium towards Dr. Friedman’s salary. Within the past two years, The Epilepsy Study Consortium received payments for research services performed by Dr. Friedman from:  Alterity, Baergic, Biogen, BioXcell, Cerevel, Cerebral, Jannsen, Lundbeck, Neurocrine, SK Life Science, and Xenon. He has also served as a paid consultant for Neurelis Pharmaceuticals and Receptor Life Sciences. He has received research support from NINDS, CDC, Epitel, and Neuropace unrelated to this study. He holds equity interests in Neuroview Technology. He received royalty income from Oxford University Press. 

Sasha Devore receives salary support from the National Institutes of Health, Department of Defense, and the Templeton World Charity Foundation unrelated to this study.

Orrin Devinsky receives grant support from NINDS, NIMH, MURI, CDC and NSF.  He has equity and/or compensation from the following companies: Tilray, Receptor Life Sciences, Qstate Biosciences, Tevard, Empatica, Engage, Egg Rock/Papa & Barkley, Rettco, SilverSpike, and California Cannabis Enterprises (CCE). He has received consulting fees from Zogenix, Ultragenyx, BridgeBio, and Marinus.  He holds patents for the use of cannabidiol in treating neurological disorders but these are owned by GW Pharmaceuticals and he has waived any financial interests.  He holds other patents in molecular biology.]

6. Please ensure that you refer to Figure 5 in your text as, if accepted, production will need this reference to link the reader to the figure.

7. Please include a copy of Table 2 which you refer to in your text on page 23.

Reviewers' comments:

Reviewer's Responses to Questions

**Comments to the Author**

1. Is the manuscript technically sound, and do the data support the conclusions?

Reviewer #1: Yes

Reviewer #2: Yes

Reviewer #3: Partly

2. Has the statistical analysis been performed appropriately and rigorously? 

Reviewer #1: Yes

Reviewer #2: Yes

Reviewer #3: Yes

3. Have the authors made all data underlying the findings in their manuscript fully available?

Reviewer #1: Yes

Reviewer #2: Yes

Reviewer #3: Yes

4. Is the manuscript presented in an intelligible fashion and written in standard English?

Reviewer #1: Yes

Reviewer #2: Yes

Reviewer #3: Yes

5. Review Comments to the Author

Reviewer #1: The authors aimed at evaluating safety of the mTOR inhibitor everolimus in treatment resistant (failure of > 2 anti-seizure medications) TSC and FCD patients undergoing surgical resection and to assess changes in mTOR signaling and molecular pathways.

I have some queries

a) the authors should provide the range age of active patients and controls since they do not appear to be age-matched. Active patients’ mean age is > 18 years old while controls’ mean age is < 18 years old.

b)How did the authors deal with the variable levels of total S6 in their analysis, particularly among the Active group?

c) A better delineation of histopathology is needed

d) The most relevant result was that no safety concerns were found in everolimus treated patients, with no reported adverse events before, during, or after surgical resection. However, owing to the low number of recruited patients and the short treatment time the authors should soften their conclusions

e) To increase the interest of the study, the authors should discuss the correlation of their results with seizure outcome after surgery in their sample

Reviewer #2: The authors report findings from their study evaluating safety and molecular mechanisms of short-term everolimus (7 days) in TSC and FCD patients.

The study has conducted well. I would like to give a few suggestions to improve the manuscript.

1. In abstract, in conclusion sub-section, authors should avoid using "These and other findings, such as changes in coagulation system activation, should be assessed in...". I would rather stop at saying that "Our findings should be confirmed in larger studies." Talking off coagulation system proteins and others is not worthwhile here.

2. The discussion should be reduced by at least a third. Also, authors should discuss the role of phospho-S6 in development of dysplastic cortex and abnormal synatptogenesis.

Reviewer #3: Leitner et al present an interesting approach to target molecular mechanisms of everolimus treatment in patients who underwent epilepsy surgery. Whilst I find merrit in the study design the presented results are very limited unless the note that it is a 'pilot study'.

- the conclusions that can be drawn are limited

- it would be an improvement if the authors show validation of their proteomics findings besides pS6 stainings; was there any difference in glutamate receptor expression throughout the individuals or myelination?

- how was the mitochondrial status?

- these data would be helpful to point out the importance of changes in metabolites

6. PLOS authors have the option to publish the peer review history of their article (what does this mean?). If published, this will include your full peer review and any attached files.

Reviewer #1: No

Reviewer #2: **Yes: **Vishal Sondhi

Reviewer #3: No

---

## [Author Response · Author response to Decision Letter 0]

25 Mar 2022

We thank the editor and reviewers for their comments and the opportunity to revise our manuscript. 

Below we have provided responses to reviewer and editor comments in red. We thank the reviewers for these suggestions, as we believe these revisions have improved our manuscript. 

Journal Requirements:

We have confirmed the manuscript meets the style formatting requirements.

We confirm that these sections are updated and matching on the submission site.

[I have read the journal's policy and the authors of this manuscript have the following competing interests: All the authors report no conflicts of interest. 

Below are the disclosures of the three authors: 

Daniel Friedman receives salary support for consulting and clinical trial related activities performed on behalf of The Epilepsy Study Consortium, a non-profit organization. Dr. Friedman receives no personal income for these activities. NYU receives a fixed amount from the Epilepsy Study Consortium towards Dr. Friedman’s salary. Within the past two years, The Epilepsy Study Consortium received payments for research services performed by Dr. Friedman from: Alterity, Baergic, Biogen, BioXcell, Cerevel, Cerebral, Jannsen, Lundbeck, Neurocrine, SK Life Science, and Xenon. He has also served as a paid consultant for Neurelis Pharmaceuticals and Receptor Life Sciences. He has received research support from NINDS, CDC, Epitel, and Neuropace unrelated to this study. He holds equity interests in Neuroview Technology. He received royalty income from Oxford University Press. 

Sasha Devore receives salary support from the National Institutes of Health, Department of Defense, and the Templeton World Charity Foundation unrelated to this study.

Orrin Devinsky receives grant support from NINDS, NIMH, MURI, CDC and NSF. He has equity and/or compensation from the following companies: Tilray, Receptor Life Sciences, Qstate Biosciences, Tevard, Empatica, Engage, Egg Rock/Papa & Barkley, Rettco, SilverSpike, and California Cannabis Enterprises (CCE). He has received consulting fees from Zogenix, Ultragenyx, BridgeBio, and Marinus. He holds patents for the use of cannabidiol in treating neurological disorders but these are owned by GW Pharmaceuticals and he has waived any financial interests. He holds other patents in molecular biology.]

We confirm that the potential competing interests do not alter adherence to PLOS ONE policies on sharing data and materials.

We are providing the uncropped and unadjusted original blot data in the supporting information file S1_raw_images.

We will confirm this formatting at submission, including confirmation that the abstract is 300 words or less.

6. Please ensure that you refer to Figure 5 in your text as, if accepted, production will need this reference to link the reader to the figure.

We have updated the manuscript to reflect the text and corresponding figure references.

7. Please include a copy of Table 2 which you refer to in your text on page 23.

We have updated the reference to “Table 2” to “S1 Table.”

Reviewer #1: The authors aimed at evaluating safety of the mTOR inhibitor everolimus in treatment resistant (failure of > 2 anti-seizure medications) TSC and FCD patients undergoing surgical resection and to assess changes in mTOR signaling and molecular pathways.

I have some queries

a) the authors should provide the range age of active patients and controls since they do not appear to be age-matched. Active patients’ mean age is > 18 years old while controls’ mean age is < 18 years old.

The age range was included in the abstract and we note in the methods section that there was no difference in the age between the Active and Control participants (p = 0.48). We have added age range for both groups in the methods section as well. Although there was no significant difference in age between the two groups, we noted in the discussion that were limitations to the study (including the broad age range), “There were several limitations to this study. There was a low number of participants due to recruitment difficulties as a result of multiple factors (few eligible candidates, few parents considered participation, many traveling long distances), several heterogeneous variables (age, brain region, neuropathology), no baseline pre-drug blood or brain tissue analyses, short-term treatment duration of 7 days, inclusion of patients with FCD that may not be mTORopathies, and lack of genetic risk factor evaluation.” 

b)How did the authors deal with the variable levels of total S6 in their analysis, particularly among the Active group?

Total S6 levels have previously been reported as being variable in glioblastoma patients but similarities are seen when comparing western blot and histology. Further, we noted in the discussion section, “Similarities in brain phospho-S6 levels between western blot and histology was reported after rapamycin in glioblastoma, with substantial variation in intra-tumoral phospho-S6 relative to baseline surgery levels but with agreement between methods and differing expression magnitudes with more dramatic differences by western blot [16]. It has also been suggested that total S6 levels may be influenced by mTOR dependent translation, in which total S6 was reduced in 2/3 patients after rapamycin [16].” We report total S6 levels as measured by western blot, histology, and proteomics (also noted below). However, one of the main outcomes was to evaluate percent of phospho-S6 relative to total S6 levels, as has been reported in human and animal model studies to decrease in response to mTOR inhibitors rapamycin and everolimus.

c) A better delineation of histopathology is needed

Additional details have been added to the immunohistochemistry section in the methods.

d) The most relevant result was that no safety concerns were found in everolimus treated patients, with no reported adverse events before, during, or after surgical resection. However, owing to the low number of recruited patients and the short treatment time the authors should soften their conclusions

We acknowledge that this is a pilot study with some limitations.

e) To increase the interest of the study, the authors should discuss the correlation of their results with seizure outcome after surgery in their sample

Correlation of our results to any changes in seizure frequency are certainly of interest, thus we noted in the methods section, “Changes to seizure frequency were not evaluated, given the brief study duration and small participant numbers with high intra-individual variability in seizure frequency.”

Reviewer #2: The authors report findings from their study evaluating safety and molecular mechanisms of short-term everolimus (7 days) in TSC and FCD patients.

The study has conducted well. I would like to give a few suggestions to improve the manuscript.

1. In abstract, in conclusion sub-section, authors should avoid using "These and other findings, such as changes in coagulation system activation, should be assessed in...". I would rather stop at saying that "Our findings should be confirmed in larger studies." Talking off coagulation system proteins and others is not worthwhile here.

We note the evaluation of coagulation system activation in particular, as coagulation should be evaluated in the context of safety in a long-term study with a larger group, thus we think it is worth mentioning as safety was one of the primary outcomes in this pilot study.

2. The discussion should be reduced by at least a third. Also, authors should discuss the role of phospho-S6 in development of dysplastic cortex and abnormal synatptogenesis.

We note in the introduction section studies related to phospho-S6 and dysplasia, “FCD may result from somatic mosaicism with pathogenic gene variants that impact the mTOR signaling pathway, particularly FCD type II with dysmorphic neurons (6, 7). Dysmorphic and balloon neurons are found histologically in TSC and FCD cases, with high ribosomal protein S6 phosphorylation (phospho-S6) levels due to increased mTOR pathway activation (6, 8).” Regarding synaptogenesis, it is reported that mTOR activity is associated with dendrite formation, including the increased activity that occurs in long-term potentiation and is altered in epilepsy models. We have emphasized this in the discussion. Interestingly, we reported that increasing phospho-S6 levels correlated with decreased synaptic transmission via proteins involved in transport and localization (Figure 3), thus synaptic transmission proteins had higher expression levels in those cases with lower phospho-S6 (seen among several of the Active participants). We have also shortened the discussion section by about 200 words.

Reviewer #3: Leitner et al present an interesting approach to target molecular mechanisms of everolimus treatment in patients who underwent epilepsy surgery. Whilst I find merrit in the study design the presented results are very limited unless the note that it is a 'pilot study'.

- the conclusions that can be drawn are limited

The results we describe in the manuscript are indeed a smaller pilot study. We agree that follow up studies are needed in larger groups and with long-term everolimus treatment to evaluate brain molecular changes and any correlations to seizure frequency.

- it would be an improvement if the authors show validation of their proteomics findings besides pS6 stainings; was there any difference in glutamate receptor expression throughout the individuals or myelination?

We noted in the results section, “The histology quantification of total S6 (RPS6, p = 0.90, Figure 2) provides validation of the proteomics results (p = 0.94, Supplemental Table 1) with some variability in the Active group that is also seen by western blot (p = 0.97, Figure 1B).” Below is a graphical representation of the validation for RPS6. 

From proteomics, we did not identify differentially expressed glutamate receptors or myelin proteins (including detected MBP, CNP, PLP1, PLP2, MAG, MOG, MOBP), only those associated with coagulation and acute phase response activation (S1 Figure). The glutamate receptor signaling pathway (p = 1.70 x 10-5, z = 0.71; S2 Table) did include detected glutamate receptors (GRIA2, GRIA3, GRIA4, GRIN1, GRIN2B, GRM2, GRM3), however this pathway was not significantly activated. There may be increased activity at the inhibitory GRM3 receptor in the Active participants related to the increased NAAG we identified by metabolomics.

- how was the mitochondrial status?

We reported that mitochondrial cellular respiration negatively correlated to higher phospho-S6 levels in Figure 3 (the most significantly correlated module to P235/236), thus cellular respiration proteins had higher expression levels in those cases with lower phospho-S6 (seen among several of the Active participants).

- these data would be helpful to point out the importance of changes in metabolites

The altered metabolites that we identified, particularly NAAG, is not necessarily directly related to mitochondrial activity. NAAG is synthesized from NAA in the cytosol by NAAG synthetase (PMID 35163193).

---

## [Decision Letter · Decision Letter 1]

3 May 2022

Pilot Study Evaluating Everolimus Molecular Mechanisms in Tuberous Sclerosis Complex and Focal Cortical Dysplasia

PONE-D-22-02043R1

Dear Dr. Devinsky,

We’re pleased to inform you that your manuscript has been judged scientifically suitable for publication and will be formally accepted for publication once it meets all outstanding technical requirements.

Kind regards,

Emilio Russo

Academic Editor

PLOS ONE

Additional Editor Comments (optional):

Reviewers' comments:

Reviewer's Responses to Questions

**Comments to the Author**

1. If the authors have adequately addressed your comments raised in a previous round of review and you feel that this manuscript is now acceptable for publication, you may indicate that here to bypass the “Comments to the Author” section, enter your conflict of interest statement in the “Confidential to Editor” section, and submit your "Accept" recommendation.

Reviewer #1: All comments have been addressed

Reviewer #2: All comments have been addressed

Reviewer #3: All comments have been addressed

2. Is the manuscript technically sound, and do the data support the conclusions?

Reviewer #1: (No Response)

Reviewer #2: Yes

Reviewer #3: Yes

3. Has the statistical analysis been performed appropriately and rigorously? 

Reviewer #1: (No Response)

Reviewer #2: Yes

Reviewer #3: Yes

4. Have the authors made all data underlying the findings in their manuscript fully available?

Reviewer #1: (No Response)

Reviewer #2: Yes

Reviewer #3: Yes

5. Is the manuscript presented in an intelligible fashion and written in standard English?

Reviewer #1: (No Response)

Reviewer #2: Yes

Reviewer #3: Yes

6. Review Comments to the Author

Reviewer #1: (No Response)

Reviewer #2: All my queries have been addressed. The manuscript is now acceptable in current format from my side.

Reviewer #3: (No Response)

7. PLOS authors have the option to publish the peer review history of their article (what does this mean?). If published, this will include your full peer review and any attached files.

Reviewer #1: No

Reviewer #2: **Yes: **Vishal Sondhi

Reviewer #3: No

---

## [Editor Report · Acceptance letter]

9 May 2022

PONE-D-22-02043R1 

Pilot study evaluating everolimus molecular mechanisms in tuberous sclerosis complex and focal cortical dysplasia 

Dear Dr. Devinsky:

I'm pleased to inform you that your manuscript has been deemed suitable for publication in PLOS ONE. Congratulations! Your manuscript is now with our production department. 

Kind regards, 

on behalf of

Prof Emilio Russo 

Academic Editor

PLOS ONE